# FlexR: Few-shot Classification with Language Embeddings for Structured Reporting of Chest X-rays

Matthias Keicher[1]                                            MATTHIAS.KEICHER@TUM.DE
Kamilia Zaripova[1]                                            KAMILIA.ZARIPOVA@TUM.DE
Tobias Czempiel[1]                                            TOBIAS.CZEMPIEL@TUM.DE
Kristina Mach[1]                                                KRISTINA.MACH@TUM.DE
Ashkan Khakzar[1]                                            ASHKAN.KHAKZAR@TUM.DE
Nassir Navab[1]                                                NASSIR.NAVAB@TUM.DE
[1] *Computer Aided Medical Procedures, Technische Universität München, Germany*

**Editors:** Accepted for publication at MIDL 2023

## Abstract

The automation of chest X-ray reporting has garnered significant interest due to the time-consuming nature of the task. However, the clinical accuracy of free-text reports has proven challenging to quantify using natural language processing metrics, given the complexity of medical information, the variety of writing styles, and the potential for typos and inconsistencies. Structured reporting and standardized reports, on the other hand, can provide consistency and formalize the evaluation of clinical correctness. However, high-quality annotations for structured reporting are scarce. Therefore, we propose a method to predict clinical findings defined by sentences in structured reporting templates, which can be used to fill such templates. The approach involves training a contrastive language-image model using chest X-rays and related free-text radiological reports, then creating textual prompts for each structured finding and optimizing a classifier to predict clinical findings in the medical image. Results show that even with limited image-level annotations for training, the method can accomplish the structured reporting tasks of severity assessment of cardiomegaly and localizing pathologies in chest X-rays.

**Keywords:** Structured report generation, Contrastive language-image pretraining, Few-shot classification, Chest X-ray diagnosis

## 1. Introduction

Radiologists often spend a significant amount of time on documentation and report-writing, rather than focusing on individual patient needs. Therefore, structured reporting is highly valued in the field of radiology for not only saving valuable time but also for standardizing content and terminology (Nobel et al., 2021; Hong and Kahn, 2013). According to Nobel et al. (2020), structured reporting can be defined as an IT-based method to import and arrange the medical content into the radiological report. It facilitates the generation of standardized reports with a structured representation of clinical findings. Professional radiology societies such as RSNA and ESR endorse structured reporting and standardized reports, as it simplifies communication and makes the reports machine-readable. These features are beneficial for a variety of purposes, such as quality assurance, clinical trials, and the internationalization of data.

In contrast, most deep learning methods proposed for automated reporting focus on the generation of free-text reports. However, despite their potential benefits, generated free-text reports can suffer from the same limitations as manually written free-text reports, such as a lack of standardization and, therefore, difficulties in assessing clinical accuracy (Pino et al., 2021). A significant challenge in automating structured reporting is the lack of extensive, high-quality collections of structured annotations that are publicly available. Furthermore, the lack of standardization in reporting templates across hospitals and countries adds complexity to the task. Therefore, a method is needed to use the abundance of unstructured free-text radiology reports and images available and adapt to new structured reporting templates with limited annotations.

To this end, we present FlexR, a flexible few-shot learning method for predicting fine-grained clinical findings for structured reporting. We use self-supervised pretraining on pairs of chest X-rays and free-text radiology reports to extract knowledge from large amounts of unstructured data and utilize it to predict structured findings defined by sentences of reporting templates that can easily be adjusted. Our results demonstrate that even with minimal image-level annotations, FlexR can predict the severity of cardiomegaly and localize pathologies in chest X-rays.

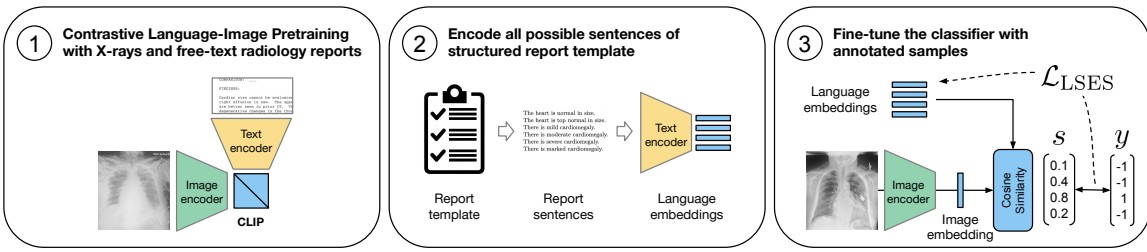

Figure 1: The Few-shot classification with Language Embeddings for chest X-ray Reporting (FlexR) method leverages self-supervised pretraining to predict fine-grained clinical findings in radiology images, using only a few high-quality annotations. The process is composed of three key steps: (1) contrastive language-image pretraining on a dataset of radiology images and unstructured reports, (2) encoding of clinical findings from structured reports, and (3) fine-tuning of the resulting language embeddings.

## 1.1. Related work

Recent advances in computer vision and natural language processing (NLP) led to significant progress in the automated diagnosis of chest X-rays and the generation of radiology reports (Wang et al., 2018; Hou et al., 2021). In contrast to the generation of free-text radiology reports, only a few works focus on automating structured reporting. Pino et al. (2021) proposed structured report generation with a classification model of high-level pathologies used to select sentences from a template. Bhalodia et al. (2021) used an object detector to localize pneumonia and predict further attributes about the lesions, which could be used in

a structured reporting setting. Similar to our work, Syeda-Mahmood et al. (2020) classified the clinical findings in X-ray images on a fine-grained level as a one-hot-encoded vector and used this to retrieve similar reports to generate free-text reports.

The concept of contrastive language-image pretraining (CLIP) (Radford et al., 2021) is to generate a joint representation of text and image pairs. This joint embedding proved to be also very useful for downstream tasks like classification and captioning. Trained originally on general data scraped from the internet, CLIP has recently been adopted to the radiology domain to improve retrieval-based radiology report generation (Endo et al., 2021). Boecking et al. (2022) further improved this pretraining by using semantic concepts and discourse characteristics. Wang et al. (2022) added pretraining on unpaired datasets to CLIP and added a semantic matching loss. The generation of language-image representations, without CLIP, has also been investigated for the report generation task using weakly-supervised contrastive pretraining (Yan et al., 2021), joint embeddings for pulmonary edema assessment (Chauhan et al., 2020), mutual information maximization for chest X-ray classification (Liao et al., 2021), and attention-based contrastive learning (Huang et al., 2021)

Few-shot learning in medical imaging aims to overcome the challenge of obtaining large amounts of accurate labels by only requiring few annotated images. For the diagnosis of chest X-rays, Paul et al. (2021) employed a discriminative autoencoder ensemble in a few-shot setting and (Jia et al., 2020) explored the few-shot generation of reports for rare diseases. Recently, methods that use CLIP have also been evaluated for zero-shot prediction using only text embeddings of pathologies with promising results. Both Seibold et al. (2022) and Tiu et al. (2022) perform a zero-shot classification using language prompts of diseases and their negation. Huang et al. (2021) and Boecking et al. (2022) showed the effectiveness of their improved pretraining in a zero- and few-shot setting in predicting chest pathologies. In contrast to our work, these works focus on classifying the multi-label presence of pathologies and do not predict fine-grained labels like disease localization or grading.

## 2. Method

Our Few-shot classification with Language Embeddings for chest X-ray Reporting (FlexR) method uses self-supervised pretraining to predict clinical findings defined by text prompts for a given radiology image. Our method's primary goal is to use large amounts of unstructured radiology data to predict structured, fine-grained clinical findings with only a few high-quality annotations. Specifically, we extract all possible sentences of a structured radiology report template and define these as possible clinical findings. Next, we project them onto a joint language-image embedding space and use these language embeddings of clinical findings to predict the ones most similar to the input image embedding. As shown in Figure 1, our method consists of three steps:

**1. Contrastive language-image pretraining (CLIP)** with unlabeled pairs of radiology reports and images: Due to the domain gap, the CLIP model must be retrained on a dataset with radiological images and reports to be applied in our approach.

**2. Language embeddings of clinical findings:** In the second step, we extract all possible options for the structured reporting template as individual sentences and encode

them with the CLIP text encoder, resulting in a text embedding $T_i$ for each clinical finding. In the example of detecting and grading cardiomegaly in chest radiographs, they could be expressed by the prompts in Table 1 like *There is mild cardiomegaly.*

**3. Fine-tuning the classifier:** The result of our approach, the classifier consists of a CLIP image encoder to obtain the embedding $I_i$ of the input image, as well as the embeddings $W = \{T_1, T_2, \ldots, T_C\}$ initialized by the text embeddings of clinical findings, where $C$ is the number of clinical findings defined by the structured reporting template. To classify the input image, we calculate the cosine similarity $s = \{s_1, s_2, \ldots, s_C\}$ between $I_i$ and each clinical finding in $W$.

We use the fact that many textual prompts share common information with other sentences from similar parts of the template, providing a useful clustering of medically similar findings and label dependencies. For example, *lung opacity in the left lung* and *lung opacity in the upper left lung* have almost identical language embeddings. We even noticed that, in rare cases, two different prompts have the same embedding using the initialization $W$. Therefore, we propose to optimize the clinical finding embeddings in $W$ and the image encoder using the Log-Sum-Exp Sign loss to ensure that different prompts result in different language embeddings.

We propose using the Log-Sum-Exp Sign (LSES) loss function (Jin et al., 2021) to optimize the clinical finding embeddings initialized by the text encoder. The labels of each clinical finding are defined as $y = \{y_1, y_2, \ldots, y_C\}$ with $y_i \in \{1, -1\}$, representing the presence and absence of a clinical finding in the report, respectively. Given the cosine similarity $s$ between image and finding embedding, we define the $\mathcal{L}_{\text{LSES}}$ loss as $\mathcal{L}_{\text{LSES}} = \log\left(1 + \sum_{i=1}^{C} e^{-y_i \gamma s_i}\right)$. which inherently assigns a higher weight to misclassified classes while leaving the embeddings of correctly initialized classes largely unchanged. This effect can be adjusted with the hyperparameter $\gamma$, which further increases the loss for misclassified embeddings and decreases it for good embeddings. This mechanism is effective in classification tasks with a long-tailed distribution like human-object interaction recognition (Jin et al., 2021) and is therefore suited for the long-tailed distribution in structured reporting. In $\mathcal{L}_{\text{LSES}}$, 1 is added to the summands to give the loss a lower bound of 0.

## 3. Experimental setup

In this section, we detail the experimental setup and the dataset employed in our study. We conduct domain-specific contrastive pretraining and evaluate the performance of FlexR on two structured reporting tasks: the assessment of cardiomegaly severity and the localization of pathologies in chest X-rays. For more information, please refer to the appendix.

### 3.1. Dataset

We use the MIMIC-CXR-JPG v2.0.0 (Johnson et al., 2019a) dataset, which is derived from the MIMIC-CXR dataset consisting of 377,110 chest radiographs associated with 227,827 imaging studies and free-text reports (Johnson et al., 2019b; Goldberger et al., 2000). The labels for structured reports are extracted from the medical scene graph dataset Chest ImaGenome (Wu et al., 2021) consisting of 242072 anatomy-centered scene graphs for the

Table 1: Options for cardiomegaly severity in the *Chest Xray - 2 Views* RadReport template with the total amount of labels extracted from MIMIC-CXR reports.

| Severity | Initialization prompt | Training | Validation | Testing |
|---|---|---|---|---|
| Normal | The heart is normal in size. | 3140 | 478 | 943 |
| Top Normal | The heart is top normal in size. | 635 | 72 | 160 |
| Mild | There is mild cardiomegaly. | 6084 | 809 | 1816 |
| Moderate | There is moderate cardiomegaly. | 8696 | 1164 | 2619 |
| Severe | There is severe cardiomegaly. | 2231 | 335 | 676 |
| Marked | There is marked cardiomegaly. | 246 | 36 | 85 |
| | | 21032 | 2894 | 6299 |

MIMIC-CXR image data. It provides 1256 combinations of relation annotations between 29 anatomical locations and their attributes.

### 3.2. Domain-specific contrastive language-image pretraining

For contrastive pretraining, we use all reports and images in the MIMIC-CXR training set. We observed severe overfitting when fine-tuning the original CLIP model with a ViT-16/B backbone (Dosovitskiy et al., 2021) using the original CLIP tokenizer and text encoder. Therefore, we replaced the image and text encoders with domain-specific pretrained encoders DenseNet121 (DN121) (Huang et al., 2017) pretrained on the detection of pathologies and SciBERT (SB) (Beltagy et al., 2019) pretrained on a large corpus of scientific text. Contrastive pretraining with these encoders showed better generalization, and we, therefore, used this model for all following experiments and compared it with the unmodified ViT-B/16-CLIP model initially fine-tuned. The appendix includes more pretraining details.

### 3.3. Few-shot classification of clinical findings

In the main set of experiments, we investigate the effectiveness of FlexR in using knowledge extracted with contrastive pretraining for the few-shot classification of fine-grained clinical findings. We do this by intentionally decreasing the amount of training data and evaluating the models in an N-shot setting, where N-shot refers to using N annotated samples per class. Finally, we compare the results to an upper bound using all available annotated data.

#### 3.3.1. SEVERITY PREDICTION OF CARDIOMEGALY

The first few-shot experiment evaluated the model's ability to adapt to a new structured reporting workflow defined by an existing reporting template. For this, we chose the assessment of cardiomegaly severity in the TLAP-endorsed structured reporting template "Chest Xray - 2 Views"[1]. The exact sentences from this template were used as language embeddings, and the associated labels were extracted from MIMIC-CXR using simple keyword matching. Table 1 shows the prompts and their distribution. The methods were evaluated using the area under the receiver operating characteristic curve (AUC).

---

1. created by Penn Medicine: https://radreport.org/home/144/2011-10-21%2000:00:00

### 3.3.2. Localized pathology detection

Due to the lack of publicly available reporting templates with annotations, we model the localization of pathologies in chest radiographs as a surrogate reporting task. This task was introduced by AnaXNet Agu et al. (2021), and ImaGenome (Wu et al., 2021), providing annotations for MIMIC-CXR as graphs. The 9 pathologies to be detected and localized are: *Lung Opacity*, *Pleural Effusion*, *Atelectasis Enlarged Cardiac Silhouette*, *Pulmonary Edema/Hazy Opacity*, *Pneumothorax*, *Consolidation*, *Fluid Overload/Heart Failure* and *Pneumonia*. The dataset contains 19 anatomical locations, including different regions of the lung, hilar structures, costophrenic angle, and mediastinum, as well as the cardiac silhouette and trachea. For each patient, we extract the triplet of *pathology* located in the *anatomical site* from the graph provided. For all patients, this resulted in 98 (of 162 possible) unique combinations of pathology and location. By joining the *pathology* and *location* with *"in the"*, we create the template sentences used as an initialization of the classifier, for example, *"Consolidation in the left lung"*. Following Agu et al. (2021), we calculate the AUC for all possible locations of each pathology and average them to one location-sensitive AUC per pathology. The pathology localization task was used in the ablation study since there are more annotations available.

## 4. Results

In this section, we present an ablation study of FlexR and the results in two few-shot tasks.

### 4.1. Ablation study and few-shot severity grading of cardiomegaly

The ablation study in Table 2 on the task of localizing pathologies shows that the FlexR model outperforms the random initialization without language embeddings and the ViT-B/16-CLIP backbone fine-tuned on MIMIC-CXR. This confirms the better generalization of the domain-adapted CLIP model and that the initialization with language embeddings improves the few-shot performance. FlexR outperforms the naïve transfer learning baseline from a DenseNet121 pretrained on detecting non-localized pathologies. With an increasing number of samples seen per class, the gap between FlexR and the other models decreases. The second part of Table 2 shows the results of detecting and grading cardiomegaly with the language embeddings extracted from a real-world RadReport reporting template. Over the naïve transfer learning baseline FlexR has an increase in AUC of 0.06 in the 1-shot case and 0.07 in the 5-shot learning over the naïve transfer learning baseline. Another interesting finding is that the DenseNet121 optimized by cross-entropy required oversampling of underrepresented classes to gain a similar performance as FlexR. Without oversampling, FlexR reached its best AUC of 0.82 when trained on all data. This could be explained by the inherent class weighting in the LSES-loss used by FlexR.

### 4.2. Few-shot detection and localization of pathologies

In the few-shot setting of localized pathology detection, FlexR showed the best result with a 0.07 higher AUC for 1-shot learning and a 0.08 increase for 5-shot learning compared to transfer learning with a pretrained DenseNet121. In Table 3 FlexR is compared with global classification baselines and methods based on object detection that use the full training

Table 2: Ablation of the FlexR method with different backbones and weight initializations on the task of pathology localization and results for cardiomegaly grading in comparison with naïve transfer learning. Mean AUC is used as the evaluation metric. N-shot refers to N annotated samples per class used for training. Our proposed approach is highlighted in bold.

| Method | Backbone | Pretraining | 1-shot | 5-shot | 10-shot | 100-shot | sampled | all |
|---|---|---|---|---|---|---|---|---|
| *Ablation on localizing pathologies* | | | | | | | | |
| MLP | DN121 | pathologies | 0.67 | 0.69 | 0.71 | 0.76 | 0.77 | 0.84 |
| FlexR | ViT-B/16-CLIP | CLIP | 0.66 | 0.70 | 0.73 | 0.75 | 0.77 | - |
| FlexR | DN121+SB | random init. | 0.67 | 0.72 | 0.75 | 0.79 | 0.81 | - |
| **FlexR** | **DN121+SB** | **CLIP** | 0.74 | 0.77 | 0.78 | 0.80 | 0.81 | 0.84 |
| *Grading task: Cardiomegaly severity prediction* | | | | | | | | |
| MLP | DenseNet121 | pathologies | 0.59 | 0.65 | 0.68 | 0.75 | 0.79 | 0.73 |
| **FlexR** | **DN121+SB** | **CLIP** | 0.65 | 0.72 | 0.74 | 0.77 | 0.78 | 0.82 |

Table 3: Comparison of FlexR against baselines using all available data with and without localization of pathologies as well as naïve transfer learning in the few-shot setting. AUC is used as the evaluation metric. N-shot refers to N annotated samples per class used for training. The proposed method is marked bold.

| Method | Lung Opac. | Pleural Eff. | Atelectasis | Enl. Card. S. | Pulm. Edema | Pneumothor. | Consolidation | Heart Failure | Pneumonia | **Avg. AUC** |
|---|---|---|---|---|---|---|---|---|---|---|
| *Multi-label classification with no localization on global view using all data* | | | | | | | | | | |
| DenseNet169 (Agu et al., 2021) | 0.91 | 0.94 | 0.86 | 0.92 | 0.92 | 0.93 | 0.86 | 0.87 | 0.84 | 0.89 |
| DenseNet169 | 0.87 | 0.90 | 0.79 | 0.86 | 0.85 | 0.83 | 0.75 | 0.77 | 0.75 | 0.82 |
| DenseNet121 | 0.88 | 0.91 | 0.81 | 0.87 | 0.87 | 0.87 | 0.79 | 0.80 | 0.77 | 0.84 |
| ViT-B16 | 0.88 | 0.91 | 0.80 | 0.87 | 0.86 | 0.85 | 0.77 | 0.78 | 0.76 | 0.83 |
| *Fully supervised object detection with bounding boxes and high-resolution crops using all data* | | | | | | | | | | |
| FasterR-CNN (Agu et al., 2021) | 0.84 | 0.89 | 0.77 | 0.85 | 0.87 | 0.77 | 0.75 | 0.81 | 0.71 | 0.80 |
| AnaXNet (Agu et al., 2021) | 0.88 | 0.96 | 0.92 | 0.99 | 0.95 | 0.80 | 0.89 | 0.98 | 0.97 | 0.93 |
| ***Few-shot**, detector-free localization on global view ($224 \times 224$)* | | | | | | | | | | |
| DenseNet121 1-shot | 0.70 | 0.76 | 0.64 | 0.77 | 0.70 | 0.60 | 0.66 | 0.62 | 0.58 | 0.67 |
| DenseNet121 5-shot | 0.72 | 0.78 | 0.66 | 0.78 | 0.73 | 0.64 | 0.67 | 0.64 | 0.62 | 0.69 |
| DenseNet121 (all data) | 0.83 | 0.89 | 0.79 | 0.87 | 0.84 | 0.89 | 0.83 | 0.81 | 0.82 | 0.84 |
| **FlexR 1-shot** | 0.72 | 0.83 | 0.69 | 0.82 | 0.77 | 0.72 | 0.74 | 0.73 | 0.67 | 0.74 |
| **FlexR 5-shot** | 0.75 | 0.84 | 0.71 | 0.82 | 0.79 | 0.78 | 0.76 | 0.73 | 0.71 | 0.77 |
| FlexR (all data) | 0.82 | 0.89 | 0.78 | 0.87 | 0.84 | 0.90 | 0.83 | 0.80 | 0.81 | 0.84 |

data. To establish an upper-bound AUC, we report the image encoders used for global pathology detection without the localization of diseases. Compared to object detection methods, the few-shot methods expectedly do not reach the AUC of AnaXNet, which uses all training data, is fully supervised with bounding boxes, and refines features extracted from high-resolution crops.

## 5. Discussion

Modeling radiology report generation as a classification task has the benefit of allowing for a direct evaluation of the clinical correctness of reports as opposed to an evaluation with NLP metrics. Also, it aligns with the trend in radiology to embrace structured reporting and standardized reports. However, to capture the nuances of radiology reports, the classification of clinical findings has to be highly fine-grained, and while there is an abundance of unstructured data like free-text reports, detailed, structured, high-quality annotations are scarce. We, therefore, motivated the need and introduced a method to make use of this unstructured data and learn to predict fine-grained clinical findings given only a few annotated samples per class. Our results show that our method can easily be adapted to a hospital-specific reporting template and outperforms the baseline in the severity assessment of cardiomegaly and the localization of pathologies in chest X-rays in a few-shot setting.

Nevertheless, self-supervised pretraining with CLIP and therefore FlexR is only viable if large amounts of domain-specific image-text pairs are available. This might not be the case with other medical applications. At the same time, using all data outperformed few-shot learning, so if possible, all labels should be used for training. Specifically, the results indicate that the localization of diseases in chest X-rays is better suited for a specialized object-detection-based model using full supervision with bounding boxes. In contrast to Seibold et al. (2022) and Tiu et al. (2022), our approach utilizes only a single negative prompt, a healthy patient without any findings, rather than negative prompts for every pathology. Adapting this strategy could potentially improve performance in detecting diseases by having two embeddings as a reference. Furthermore, label dependencies (e.g., located in the *left lung* and *lower left lung*) have only been modeled implicitly in FlexR by the similarity of text in prompts, which could, in the future, be explicitly modeled. Ultimately, the greatest potential for enhancement lies in refining the pretraining of joint vision-language representations. This objective may be accomplished through the utilization of additional data or by improving the methodology to tackle the severe class imbalance of clinical findings.

To allow a more extensive evaluation of structured reporting in the future, a publicly available, standardized reporting template for chest X-rays with a high level of detail and corresponding annotations for datasets like MIMIC-CXR are needed. While Wu et al. (2021) and Jain et al. (2021) provide highly detailed, structured annotations in the form of graphs, a translation to a real-life reporting template and a benchmark for comprehensive, structured reporting would facilitate future research in this direction.

## 6. Conclusion

In this paper, we highlight the need for methods for structured reporting because standardized reports can formalize the evaluation of knowledge embedded in the representations

of neural networks. We proposed a Few-shot classification with Language Embeddings for chest X-ray Reporting (FlexR) method that utilizes self-supervised pretraining with CLIP to predict fine-grained clinical findings for a given radiology image. Our results showed that even with limited image-level annotations, our method can predict the structured reporting subtasks of cardiomegaly severity assessment and localizing pathologies in chest X-rays in a few-shot setting.

## Acknowledgments

The authors acknowledge the financial support by the Federal Ministry of Education and Research of Germany (BMBF) under project DIVA (FKZ 13GW0469C). Ashkan Khakzar was partially supported by the Munich Center for Machine Learning (MCML) with funding from the BMBF under project 01IS18036B. Kamilia Zaripova and Kristina Mach were partially supported by the Linde & Munich Data Science Institute, Technical University of Munich Ph.D. Fellowship.

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

## Appendix A. Log-Sum-Exp used in the LSES loss

The Log-Sum-Exp (LSE) function defined as

$$LSE = \log \left( \sum_{i=1}^{C} e^{x_i} \right) \tag{1}$$

is a smooth approximation of the maximum function $max\{x_1, x_2, \ldots, x_i\}$ with softmax being its derivative. With $x_i = -y_i \gamma s_i$ it assigns the highest loss to classes that are present in the report but have a low similarity with the image embedding or have a high similarity but are not present. At the same time, the softmax gradient keeps the correctly initialized class weights stable. More details can be found in the work of Jin et al. (2021) that served as an inspiration for the use of the loss.

## Appendix B. Implementation details

### B.1. Data processing

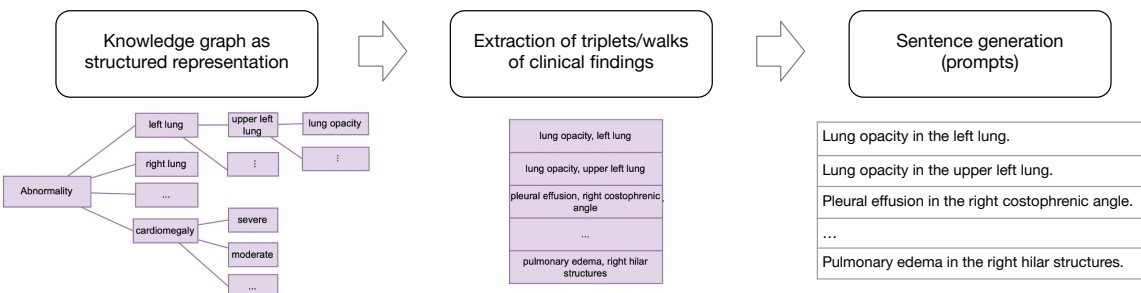

Figure 2: Extraction of natural language prompts from the ImaGenome (Wu et al., 2021) knowledge graph representing sentences in a structured reporting template.

We use the data split provided by ImaGenome with Posterior-Anterior (PA) and Anterior-Posterior (AP) radiographs resulting in 166512 training images, 23952 validation images, and 47389 test images after preprocessing. The images were processed with MONAI 0.8.0 and the dataloader is implemented with ffcv 0.0.2. All images are resized to 224x224, padded if needed, and scaled to the range [-1,1]. For training, the following image augmentations were applied: random crop with at least 75% image size, random rotation up to $\pm 15°$, and a color jitter of 10% brightness as well as 20% contrast and saturation. The reports were augmented by randomly sampling a sentence containing a finding in the ImaGenome scene graph. For healthy patients with no findings, a random sentence from the full report was randomly sampled.

### B.2. FlexR Model - Cosine similarity

The cosine similarity of the input image embedding and the language embeddings of the clinical finding prompts is implemented with a fully connected layer without bias. Both

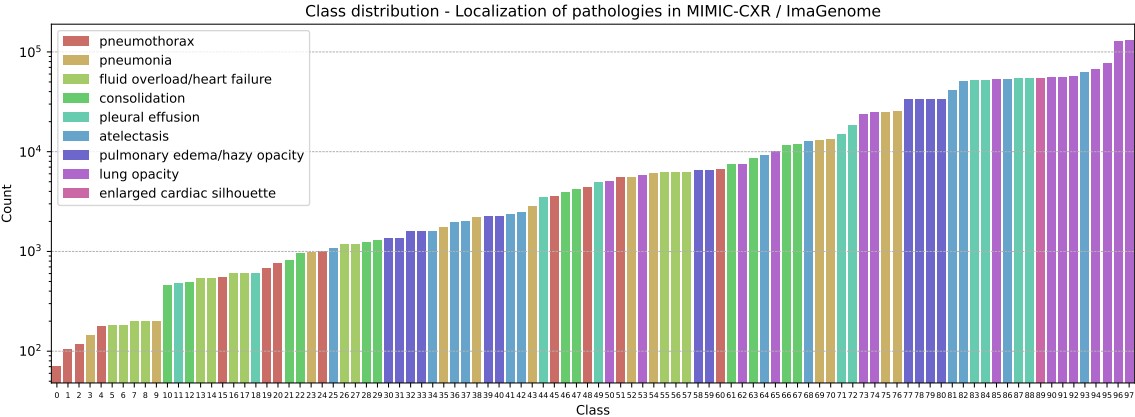

Figure 3: Long-tailed distribution of classes in the MIMIC-CXR / ImaGenome task of localizing nine pathologies in 19 anatomical locations. Sorted by ascending frequency of class occurrence and plotted on a logarithmic scale.

the initial text embeddings and the image embeddings are normalized and the linear layer calculates the dot product between the embeddings while allowing the embeddings of the clinical findings to be differentiable. Algorithm 1 shows the pseudo-code of a PyTorch implementation.

---

**Algorithm 1:** Prompt Similarity Pytorch Module

---

**class** *PromptSimilarity*(nn.Linear)

    **method** \_\_*init*\_\_(prompt_embeddings)

        out_features, in_features ← prompt_embeddings.shape;

        **call** super().\_\_**init**\_\_(in_features, out_features, bias=False);

        self.weight.data ← F.normalize(prompt_embeddings);

    **method** *forward(x)*

        x ← F.normalize(x);

        x ← super().**forward**(x);

        **return** x;

---

### B.3. Implementation and training details

All networks were trained with PyTorch 1.10 and PyTorch lightning 1.5.10 in native mixed precision. The transformer-based models and tokenizers are implemented using the huggingface library (transformers 4.16.2). All classification models were trained on a single NVIDIA A40. The DenseNet and Vision Transformer classifiers were trained for 25 epochs with the same hyperparameters as in (Agu et al., 2021): Adam optimizer, a learning rate of 1e-4 and unweighted binary cross-entropy loss. The FlexR models were fine-tuned for 10 epochs with a learning rate of 1e-4, AdamW optimizer with no weight decay, and learning

rate scheduler with cosine annealing decay and 1 epoch linear warmup. During fine-tuning, all weights are optimized including the image encoder. Fine-tuning only the language embeddings and zero-shot inference did not yield useful results and was therefore omitted from the experiments. The model performing best on the validation set was used for testing. After a hyperparameter search on the localized pathology task with 50, 100, and 150 the $\gamma$ parameter of the LSES loss was set to 50 for all experiments. For few-shot learning, an epoch is defined as seeing 128 images per class. In the setup of sampling all classes, the same amount of images was chosen per class to ensure comparability while having access to the entire dataset. A batch size of 256 is used for the classification baselines and FlexR finetuning. All experiments were repeated 5 times with different seeds and the average results are reported. The DenseNet121 baseline for cardiomegaly severity assessment is trained with a cross-entropy loss. To allow a fair comparison with our custom CLIP model described below, the DenseNet121 used as a baseline in the two few-shot tasks was also initialized with a model pretrained on detecting pathologies in MIMIC-CXR without localization.

### B.3.1. CONTRASTIVE PRETRAINING DETAILS

The CLIP models were fine-tuned on all radiology reports and chest X-ray images in the MIMIC-CXR training dataset with 8 NVIDIA A40 for 300 epochs with a batch size of 128 using an AdamW optimizer with a learning rate of 5e-6, no weight decay on the normalization layer, and bias and 0.1 weight decay on all other parameters. The learning rate was decayed with a single cosine annealing schedule and 1 epoch linear warmup. The huggingface library (transformers 4.16.2) was used for the implementation of vision-language models. Initially we fine-tuned the weights provided by OpenAI for the ViT-B/16-CLI configuration (https://huggingface.co/openai/clip-vit-base-patch16). After observing overfitting on the MIMIC-CXR dataset, we replaced the image encoder with a DenseNet121 and the language encoder with SciBERT (https://huggingface.co/allenai/scibert_scivocab_uncased). The DenseNet121 was pretrained with the detection of pathologies without localization (see section 3.3.2) on the MIMIC-CXR training set. Since the embedding dimensions of the image and text encoder might be different their embeddings are projected to a joint embedding size of 512 with a linear layer. Following the original CLIP paper (Radford et al., 2021) and its implementation in huggingface, the cosine similarity between the text and image embeddings was calculated and a contrastive loss applied to encourage similarity between corresponding pairs of text and image inputs while pushing apart non-corresponding pairs. Specifically, the contrastive loss is a cross-entropy loss applied on both the rows and columns of the similarity matrix, and the average of these two losses is used for backpropagation.

