# OpenReview forum: "FlexR: Few-shot Classification with Language Embeddings for Structured Reporting of Chest X-rays"
_MIDL.io/2023/Conference — MIDL 2023 Poster_

### Official Review · Reviewer_Xbq6 · 2023-01-29

**Confidence:** 5
**Preliminary Rating:** 3
**Recommendation:** Poster

**Summary:**

The key idea of this paper is to use a contrastive language-image model to predict clinical findings defined by sentences in structured reporting templates. This method involves training the model using CXR and related free-text radiological reports, then creating textual prompts for each structured finding and optimizing a classifier to predict clinical findings in the medical image. The experiments conducted showed that even with limited image-level annotations for training, the method can still be used for assessments of cardiomegaly and locating pathologies in chest X-rays.

**Strengths:**

The strengths of this paper are that it provides a method to automate CXR reporting while maintaining accuracy. The approach involves training a contrastive language-image model using chest X-rays and related free-text radiological reports, then creating textual prompts for each structured finding and optimizing a classifier to predict clinical findings in the medical image. This research has potential value because it can provide consistency when evaluating clinical correctness, as well as save time by automating the process of generating reports. On the basis of its scientific merits and potential value to the community, I would rate this paper highly.


**Weaknesses:**

- One of the weaknesses of this paper is that it does not provide a comprehensive evaluation of its proposed method. The experiments conducted only showed that the method can be used for assessments of cardiomegaly and locating pathologies in CXR, but did not evaluate other potential applications or scenarios. Additionally, since this is a validation paper, there was no discussion on how to improve upon the current approach or what further research could be done to expand upon these findings.
- Some clarifications on how exactly the contrastive language-image model is trained and optimized would help readers better understand this process.
- a discussion of how useful it is to try and frame radiologic reporting into a classification task
- I would have liked to see more about the idea that is hinted at the beginning of the paper and to use these method to help and define structured reporting for AI-supported radiologists.


**Deanonymize Review:**

no

**Detailed Comments:**

suggestions for further references:
Wang X, Peng Y, Lu L, Lu Z, Summers RM. Tienet: Text-image embedding network for common thorax disease classification and reporting in chest x-rays. InProceedings of the IEEE conference on computer vision and pattern recognition 2018 (pp. 9049-9058).
Hou B, Kaissis G, Summers RM, Kainz B. Ratchet: Medical transformer for chest x-ray diagnosis and reporting. InMedical Image Computing and Computer Assisted Intervention–MICCAI 2021: 24th International Conference, Strasbourg, France, September 27–October 1, 2021, Proceedings, Part VII 24 2021 (pp. 293-303). Springer International Publishing.

typos:
p3: pertaining -> pretaining

**Paper Type:**

validation/application paper

**Questions To Address In The Rebuttal:**

In their rebuttal, I would like the authors to address how they plan to improve upon the current approach and what further research could be done to expand upon these findings. Additionally, providing a more comprehensive evaluation of their proposed method and discussing potential applications or scenarios for which it can be used would help readers better understand its capabilities.

---

### Official Review · Reviewer_AiDw · 2023-02-03

**Confidence:** 4
**Preliminary Rating:** 4

**Summary:**

This paper tackles of a problem of predicting clinical findings in structured reporting templates, which has the limited high-quality annotations. The proposed FlexR contains 1. CLIP retraining on X-rays and free-text radiology reports; 2. Encoding of clinical findings to structured reporting; 3. Fine-tuning for classification. LSES loss function is used to ensure the distinguish of different prompts and alleviated the long-tailed distribution issue. The proposed method is evaluated on MIMIC-CXR dataset. First, the evaluation is on the severity prediction of cardiomegaly. Second, the evaluation is on pathology detection.

**Strengths:**

-This is an easy-to-follow & well-written paper. The authors give a clear introduction of the problem and conduct a comprehensive review of existing works. The methods part is clear and easy to follow.
-This paper tackles the problem of limited annotation of structured reporting of X-Ray. This is a significant problem in the field.
-The application of CLIP to the field is effective.





**Weaknesses:**

- While CLIP is shown to be effective in the medical report tasks. There are existing works. The authors should emphasis the novelty of this paper and compare with existing works.
- The improvements of FlexR over the other methods (DenseNet few-shot) is incremental in Table 3.


**Deanonymize Review:**

no

**Paper Type:**

methodological development

**Questions To Address In The Rebuttal:**

-Table 2, the 1-shot means one training sample? The 100-shot and the sampled leads to similar results, can authors comment on this?
-Table 3, ViT-B16 and FlexR (all data) has similar results. Does that mean when the data all available, the CLIP based FlexR does not have a significant improvement? Also, this table does not show FlexR 100-shot. Will this have similar results as the other baselines? Can authors comments more ?
-Minor point: is the model trained as BioViL? (if not, why mentioning on section 2.1?)

---

### Official Review · Reviewer_UTzQ · 2023-02-03

**Confidence:** 2
**Preliminary Rating:** 2

**Summary:**

The authors present a method to generate structured findings from radiology images using unstructured data for training. From a set of images and radiology reports, they learn a joint representation that they use to encode the sentences of a structured report. Then, they fine tune the classifier with a small set of coupled images and structured report samples. The method is tested in two applications: the detection of the degree of cardiomegaly in chest x-rays and the detection of pathologies.

**Strengths:**

This is interesting work since there is a wealth of unstructured data available and high quality structured data is scarce. The method seems comprehensive. The results are fairly convincing, although the AUCs for the cardiomegaly detection and surprisingly low – maybe due to the six degrees of disease stated. What would be the normal inter-reader performance for such task?

**Weaknesses:**


The main weakness of the paper is the presentation. Although the paper seems correct, it is very hard to follow. For instance, when the authors refer to “few-shot”, I assume they refer to the use of a low amount of high quality training data. Does 1-shot refer to a single high quality training data point? With the current writing that is only an assumption on my end.


**Deanonymize Review:**

no

**Detailed Comments:**

“pertaining” - pretraining

**Paper Type:**

methodological development

**Questions To Address In The Rebuttal:**

I would like the authors to calmly think about how to improve the readability of the paper. Even after two reads it is unclear to me how the data is used. For instance, table 1 shows the number of prompts used for initializing the classifier in the second task, the prediction of the severity of cardiomegaly, however, such task is always referred as the first task. Then they state the number of training, validation and test sentences they have, I am very confused at this point, as I though the authors had only a very limited set of high-quality structured data, and I am seeing more than 30000 structured sentences.

---

### Meta-Review · Area_Chair_eMzb · 2023-02-25

**Recommendation:** Accept (Poster)
**Confidence:** 4

**Metareview:**

The paper proposes a method to predict clinical findings defined by sentences in structured reporting templates using a contrastive language-image model trained on chest X-rays and related free-text radiological reports, which can be used to fill such templates and achieves promising results with limited image-level annotations for training.
The main weak point of this paper seems the writing and the presentation as the reviewers indicate.
However, it looks like the authors have robustly responded to the comments from the reviewers.
At last, the reviewers haven't changed their scores.
I would like to see if they have seriously revised their submission.
This paper is on the edge of the borderline.